# Monitoring Residual Soil Moisture and Its Association to the Long-Term Variability of Rainfall over the Upper Blue Nile Basin in Ethiopia

**Getachew Ayehu** [1,2,*] ![ID], **Tsegaye Tadesse** [3] ![ID] **and Berhan Gessesse** [1] ![ID]

[1] Remote Sensing Research and Development Department, EORC,
Ethiopian Space Science and Technology Institute, Addis Ababa P.O. Box 33679, Ethiopia;
berhang@essti.gov.et

[2] Institute of Land Administration, Bahir Dar University, Bahir Dar P.O. Box 79, Ethiopia

[3] National Drought Mitigation Center, University of Nebraska-Lincoln, Lincoln, NE 68583, USA;
ttadesse2@unl.edu

[*] Correspondence: getachewt@essti.gov.et; Tel.: +251-911-33-6972

**Abstract:** Monitoring soil moisture and its association with rainfall variability is important to comprehend the hydrological processes and to set proper agricultural water use management to maximize crop growth and productivity. In this study, the European Space Agency's Climate Change Initiative (ESA CCI) soil moisture product was applied to assess the dynamics of residual soil moisture in autumn (September to November) and its response to the long-term variability of rainfall in the Upper Blue Nile Basin (UBNB) of Ethiopia from 1992 to 2017. The basin was found to have autumn soil moisture (ASM) ranging from 0.09–0.38 $m^3/m^3$, with an average of 0.26 $m^3/m^3$. The ASM time series resulted in the coefficient of variation (CV) ranging from 2.8%–28% and classified as low-to-medium variability. In general, the monotonic trend analysis for ASM revealed that the UBNB had experienced a wetting trend for the past 26 years (1992–2017) at a rate of 0.00024 $m^3/m^3$ per year. A significant wetting trend ranging from 0.001 to 0.006 $m^3/m^3$ per year for the autumn season was found. This trend was mainly showed across the northwest region of the basin and covers about 18% of the total basin area. The spatial patterns and variability of rainfall and ASM were also found to be similar, which implies the strong relationship between rainfall and soil moisture in autumn. The spring and autumn season rainfall explained a considerable portion of ASM in the basin. The analyses also signified that the rainfall amount and distribution impacted by the topography and land cover classes of the basin showed a significant influence on the characteristics of the ASM. Further, the result verified that the behavior of ASM could be controlled by the loss of soil moisture through evapotranspiration and the gain from rainfall, although changes in rainfall were found to be the primary driver of ASM variability over the UBNB.

**Keywords:** ESA CCI; residual soil moisture; evapotranspiration; trend; rainfall variability; CHIRPS

## 1. Introduction

Soil moisture is an essential parameter to understand various processes in agriculture, hydrology, and climate [1]. In agriculture, the spatio-temporal distribution of soil moisture determines the success of crop production because the growth and productivity of crops highly depends on the sufficient amount and timing of available moisture. Notably, in developing countries such as Ethiopia, where the livelihood and economy of the country are highly dependent on rainfed agriculture [2–4], water is a distinctly scarce and valuable resource. The Upper Blue Nile Basin (UBNB) in Ethiopia receives an adequate amount of annual rainfall (> 2000 mm), the majority of the rainfall occurs during the summer

growing season [5,6]. Agriculture in the UBNB is dominated by smallholder farmers, who are unable to produce enough amount of food by a single harvest during the main rainy season to sustain their livelihoods [7,8]. However, following the harvest of main season cropping in the UBNB, a certain amount of soil water, as residual soil moisture, is left in the soil that can last up to several months [9,10]. Residual soil moisture in this study was defined as the amount of water left in the soil following the physiological maturity or harvesting of main season cropping. Given the low irrigation facilities [11–13] and meager crop production during the main season [3,14], additional cropping in the off-season using residual soil moisture could be an alternative option to increase food and feed production in the basin.

The extent of residual soil moisture available varies temporally with hydroclimate conditions, and spatially depending on biophysical factors such as topography and soil properties of the basin [15,16]. Different works of literature signify that the dynamics of soil moisture mainly instigates from local rainfall [17,18]. Some other scholars argued that the spatio-temporal dynamics of soil moisture depends on the temporal variability of both rainfall and evapotranspiration [19,20]. For example, Cheng et al. [21] revealed that the amount of soil moisture available to crops mainly depends on rainfall, and the spatial distribution of the soil moisture trend looks like that of rainfall. Similarly, Robinson et al. [22] and He et al. [23] indicated that the spatio-temporal dynamics of soil moisture is affected by rainfall variability to a more significant extent. Again, the result obtained by Jia et al. [24] suggested that compared to other climatic variables rainfall is the primary factor responsible for the trends and variability of soil moisture. Indeed, rainfall from different season has various impact on the amount and spatio-temporal distribution of soil moisture over various seasons [25]. Yang et al. [25] reported that both the current (i.e., spring rainfall) and previous rainfall (i.e., rainfall in summer, autumn, and winter remaining in the soil by its own "memory") influenced the variability of spring soil moisture with a difference in magnitude. As a result, knowledge of soil moisture and its link with rainfall is fundamental to better understand and predict the hydrological process of the basin.

On the other hand, due to a lack of continuously measured soil moisture datasets the soil moisture conditions and its association with rainfall variability poorly understood in the UBN B. Soil moisture information is not available or not being measured regularly like other climate variables such as rainfall and temperature in the basin. Such a study indeed requires a large-scale soil moisture dataset, although obtaining soil moisture observation at this scale is often a challenge [26]. In this connection, it is hard to find studies that characterize the spatial-temporal dynamics of soil moisture and its linkage with local hydro-climate conditions in Ethiopia. However, in other places a great effort was undertaken to understand soil moisture dynamics [27–29] and its relationship with other climate variables (e.g., rainfall, temperature, evapotranspiration, and solar radiation) [19,30,31] at a range of field to a global scale, although many of the previous studies depend on point-based in-situ measurements, which are commonly inadequate to perform soil moisture monitoring on larger scales [21].

In recent years, a broad range of regional and global soil moisture products with reasonable temporal and spatial resolution from different sources, such as land surface modeling [32], remote sensing [33], and data assimilation techniques [34,35] are now available. The dependability of model-based products is controlled by the feature of meteorological forcing data [36,37]. Consequently, recent studies of soil moisture and its concurred response to climate variation over higher spatial and temporal scales are largely based on remote sensing [38,39] and data assimilations techniques [21,40]. The latest version of the soil moisture product released by the European Space Agency's Climate Change Initiative (ESA CCI), which merged active and passive microwave observations, provides relatively consistent and reliable soil moisture information worldwide over the period of 1978 to 2018 [38,41,42]. The performance of ESA CCI has been extensively validated in different regions against in-situ network observations. Dorigo et al. [43] provided comprehensive reviews of these studies. Particularly, McNally et al. [44] evaluated ESA CCI soil moisture over East Africa (including Ethiopia) comparing with the Normalized Difference Vegetation Index (NDVI) and modeled soil

moisture products. The validation of the dataset has been proven to be useful in a large number of applications such as long-term soil moisture trend analysis [24,45,46] and drought monitoring [47].

Although earlier studies conducted on the variability of soil moisture provided pertinent information, the long-term trend and variability of soil moisture in the off-season were not adequately addressed. It is also unclear to what extent rainfall variability can control soil moisture dynamics in the off-season. Thus, investigating the dynamics of soil moisture in the off-season and its response to the long-term variability of rainfall is crucial to set proper agricultural water use management in terms of maximizing crop growth and productivity [27,48]. The main objective of this study is to leverage the readily available ESA CCI soil moisture product in order to investigate the dynamics of autumn (September to November) soil moisture and its linked response to the long-term variability of rainfall over the UBNB in Ethiopia. The change in evapotranspiration (ET) might considerably affect soil moisture characteristics; therefore, it is also essential to understand how the mutual effects of ET and rainfall may bring about changes in autumn soil moisture over the UBNB. This study answers the following three questions: What is the characteristic of the autumn soil moisture inter-annual variability and trend over the UBNB from 1992 to 2017? What is the response of autumn soil moisture to the long-term effects of seasonal and annual rainfall? How does soil water loss via ET and gain through rainfall at a seasonal and annual scale determine the availability of autumn soil moisture in the UBNB?

Our paper is organized into five sections. Section 2 presents the materials and methods used in proposed study. Section 3 describes the results of the study. Major findings of the study are discussed in Section 4. Finally, Section 5 addressed the main conclusions.

## 2. Materials and Methods

### 2.1. Descriptions of the Study Area

The Upper Blue Nile Basin (UBNB) is located in the northwestern part of Ethiopia (Figure 1) and contributes the major share of the Nile River's annual water flow [5,49]. The basin is described by rugged topography with elevation ranging from 490 to 4239 m a.s.l. (Figure 1). The annual climate of the basin can be divided into two (i.e., rainy and dry) seasons. The rainy season can be split into a short rainy season from February to May and a main rainy season from June to September. The dry season occurs between October and January. The mean annual temperature in the study site is about 20.4 °C. The basin receives an annual rainfall up to 2200 mm, which mostly occurs during the wet season (June to September) and is known locally as "Kiremt" [50]. However, the basin is known by considerable spatial and temporal variations in rainfall [5,51], which makes the hydrological process of the basin very complex. Despite a range of land-use systems occurring, the livelihoods of the majority of the populations in the basin are greatly reliant on rainfed agriculture.

### 2.2. Datasets

#### 2.2.1. ESA CCI Microwave Soil Moisture

The ESA CCI soil moisture product is provided by the European Space Agency. The ESA CCI dataset is a merged multi-satellite microwave soil moisture product which combines observation from active and passive sensing microwave sensing systems [38,41]. To combine these products, the datasets were first rescaled using the Global Land Data Assimilation System (GLDAS) data as a standard reference, which has a spatial resolution of 0.25° at a daily basis and represents soil moisture layers up to 10 cm [52]. The ESA CCI soil moisture dataset provides surface soil moisture information in a volumetric unit ($m^3 m^{-3}$) and was available daily from 1978 to 2018 (v04.4) with a spatial resolution of 0.25° [41]. Besides the variation in the feature of individual data sources, the reliability of ESA CCI soil moisture products could also be affected by the adapted merging methods for combining observations from different mission and retrieval algorithms [45]. However, the merged products are superior to

either the passive or active alone [41]. In this study, the latest version (v04.4) of ESA CCI soil moisture data was used for long-term trend and variability analysis of residual soil moisture in the UBNB.

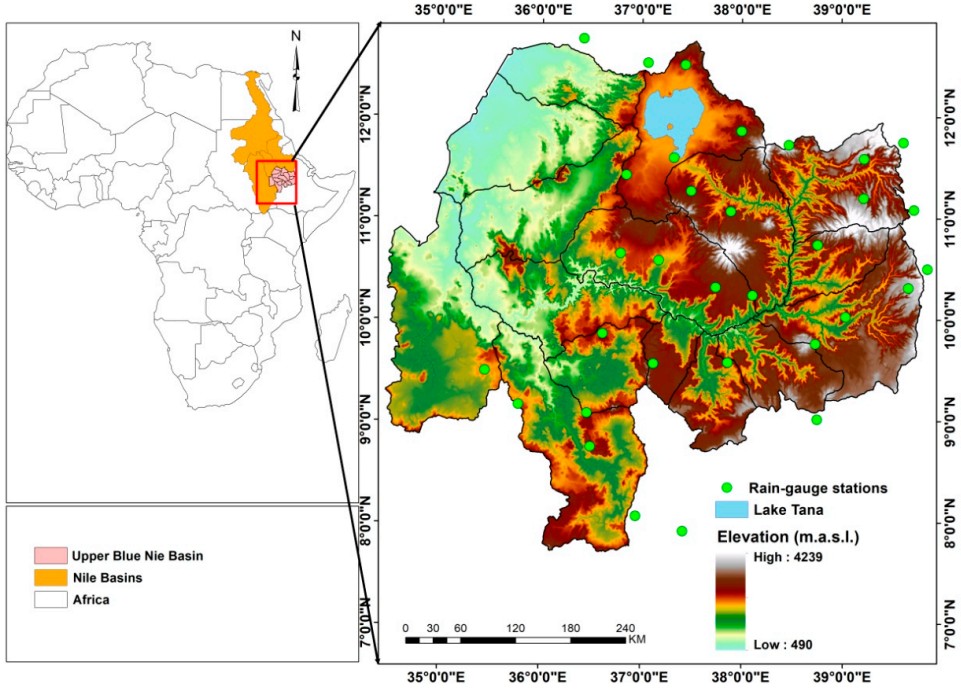

**Figure 1.** Location of the Upper Blue Nile Basin (UBNB) (Imagery source for UBNB: Shuttle Radar Topography Mission (SRTM) Global elevation data). The green bold point shows the distributions of rain gauge stations used to validate the performance of the Climate Hazards Group InfraRed Precipitation with Station (CHIRPS) satellite rainfall datasets.

### 2.2.2. FLDAS Noah Land Surface Model

The Famine Early Warning Systems Network (FEWS NET) Land Data Assimilation System (FLDAS) dataset contains a series of land surface parameters simulated from the Noah 3.6.1 model. The system generates ensembles of soil moisture, evapotranspiration (ET), and other variables based on multiple meteorological inputs or land surface models. In this study, FLDAS Noah land surface global data (Model L4) were used to extract soil moisture data (at a depth of 0–10, 10–40, and 40–100 cm depths) and ET datasets. The simulation was forced by a combination of the Modern-Era Retrospective analysis for Research and Applications version 2 (MERRA-2) data and Climate Hazards Group InfraRed Precipitation with Station (CHIRPS) 6-hourly rainfall data that was downscaled using the NASA Land Data Toolkit. The dataset provides soil moisture in volumetric units ($m^3 m^{-3}$), and ET in kilograms per meter squared per second (kg $m^{-2}$ $s^{-1}$) and is available from 1982 to present at monthly intervals with a spatial resolution of 0.10°. The dataset was obtained from the NASA Goddard Earth Science Data and Information Services Center (GES DISC) website (https://disc.gsfc.nasa.gov/). The atmospheric and land modeling communities widely use the FLDAS land surface model, and therefore, model parameters are well tested [44]. Several researchers over East Africa e.g., [53,54], used the model for various hydrological functions.

### 2.2.3. CHIRPS Rainfall Product

The CHIRPS-v2 rainfall estimate was used as a source of rainfall data in the UBNB, Ethiopia. CHIRPS is a quasi-global (500 S–500N) product provided from 1981 to near present at 0.050 spatial resolutions (~5.3 km) and at daily, pentadal (5-day), dekadal (10-day), and monthly temporal resolution (Funk et al., 2015). The dataset was created by the U.S. Geological Survey (USGS) and the Climate Hazards Group (CHG) at the University of California [55]. It integrates both ground and satellite observations to grant a global rainfall estimate with reasonably low latency, high resolution, low bias

and a long period of record. A validation study made over the UBNB revealed the great skill and immense potential of CHIRPS-v2. It can be used for various operational applications such as hydro-climate studies in areas where the gauge stations are very sparse and unevenly distributed [56]. Figure A1 (Appendix A) provides a summary of the validation results [56].

### 2.2.4. ESA CCI Land Cover

The ESA has derived a high-quality historical land cover (LC) dataset as part of their CCI program [57]. The ESA CCI LC dataset is available from 1992 to 2015 at 300m spatial resolution and with a one-year temporal interval [58]. This product was created based on ESA's Glob Cover products using the Glob Cover unsupervised classification chain. The dataset is generated using multiple sensors e.g., Systeme Probatoire d'Observation de la Terre Vegetation (SPOT-VGT), Advanced Very High-Resolution Radiometer (AVHRR), and PROBA-V with an overall accuracy of about 71% [59]. The product provides 37 LC types (Table A1, Appendix B) based on FAO LC classification systems [58]. Interested readers are referred to Plummer et al. [60]. The accuracy of the ESA CCI LC product was evaluated at a global scale [61] over Africa [62] and China [63], which can give valuable insights for specific applications. For example, Guidigan et al. [64] and Li et al. [65] used the ESA CCI LC product to study the land use and land cover changes in Benin and over the globe, respectively.

### 2.3. Methods

Since the spatio-temporal coverage of the ESA CCI soil moisture dataset is generally poor in Ethiopia before 1992, the datasets from 1992 to 2017 were selected as the study period to maintain coincident temporal coverage in the study area. Then, daily values of ESA CCI soil moisture were converted to monthly time series and averaged for September to November to construct the annual mean soil moisture for the autumn season. The same approach was applied to FLDAS NOAH soil moisture dataset. The months of September indeed belong to the wettest period in the study area; however, during this month, main season cropping reaches to the stage of physiological maturity and crops will have limited moisture intake. As a result, dry season farming may start as of September for efficient utilization of the residual soil moisture. The original unit of monthly ET is kg m$^{-2}$s$^{-1}$, which was first converted to millimeter (mm) and aggregated for each season: December to February (winter), March to May (spring), June to August (summer), and September to November (autumn). The annual series was calculated from the sum of the seasonal estimates. The same method was applied to CHIRPS rainfall values to calculate the seasonal and annual time series for the period of 1992 to 2017. For the FLDAS NOAH and CHIRPS dataset, average values of their estimates were calculated at a 0.25° grid to keep the spatial resolution consistent with ESA CCI soil moisture dataset. The datasets used in this study including their temporal and spatial resolutions are summarized in Table 1.

**Table 1.** Summary of the products used in this study.

| Product | Data Type | Spatial Cover | Temporal Cover | Spatial Reso | Temporal Reso | Sensor |
|---|---|---|---|---|---|---|
| ESA CCI | Soil moisture (m$^3$ m$^{-3}$) | Global | 1978–2018 | 0.25° | Daily | SMMR, SSM/I, TMI, AMSR-E, ASCAT, ERS |
| FLDAS Noah LSM | Soil moisture (m$^3$ m$^{-3}$) ET (kg m$^{-2}$s$^{-1}$) | Global | 1982–present | 0.10° | Monthly | MERRA-2, CHIRPS |
| CHIRPS v2 | Rainfall | Global | 1981–present | 0.05° | Daily, Pentadal, Dekadal, Monthly | NOAA TIR satellite, TRMM |
| ESA CCI | Land cover | Global | 1992–2015 | 300 m | 1 year | MERIS; FR/RR SPOT-VGT; AVHRR; PROBA-V |
| DEM | Elevation | Global | - | 30 m | - | SRTM |

### 2.3.1. Variability and Trend Analysis

Variability analysis involves the generation of the long-term mean (LTM), calculating the coefficient of variation (CV), and an anomaly. The LTM for autumn soil moisture (ASM) and rainfall (both at a seasonal and annual time scale) were calculated from 1992 to 2017. The analyses were done using "raster" packages in the R platform [66]. The CV is calculated to evaluate the spatio-temporal variation for the time series and is computed using Equation (1):

$$CV = \frac{\sigma}{\mu} \times 100 \tag{1}$$

where *CV* is the coefficient of variation; $\sigma$ is the standard deviation and $\mu$ is the mean value. The standardized anomalies of soil moisture and rainfall have been calculated to indicate the departures of each year total from the LTM, and were computed as follows (Equation (2)):

$$Z_a = \frac{\left(X_i - \overline{X}_i\right)}{s} \tag{2}$$

where $Z_a$ is standard anomaly; $X_i$ is annual value of a particular year; $\overline{X}_i$ is long-term mean (LTM) annual values, and *s* is the standard deviation over a period of observations (1992 to 2017, in our case). A negative value of $Z_a$ represents periods of below-normal value, while positive value indicates above-normal values. To further assess the variability of soil moisture across the different land cover classes and for three different decades of the UBNB, soil moisture values from 1992 to 2000 were extracted using a ESA CCI land cover map of 1995, likewise from 2001 to 2010 (2005) and 2011 to 2017 (2015), respectively. Figure 2b presents the land cover map of the UBNB for 2015.

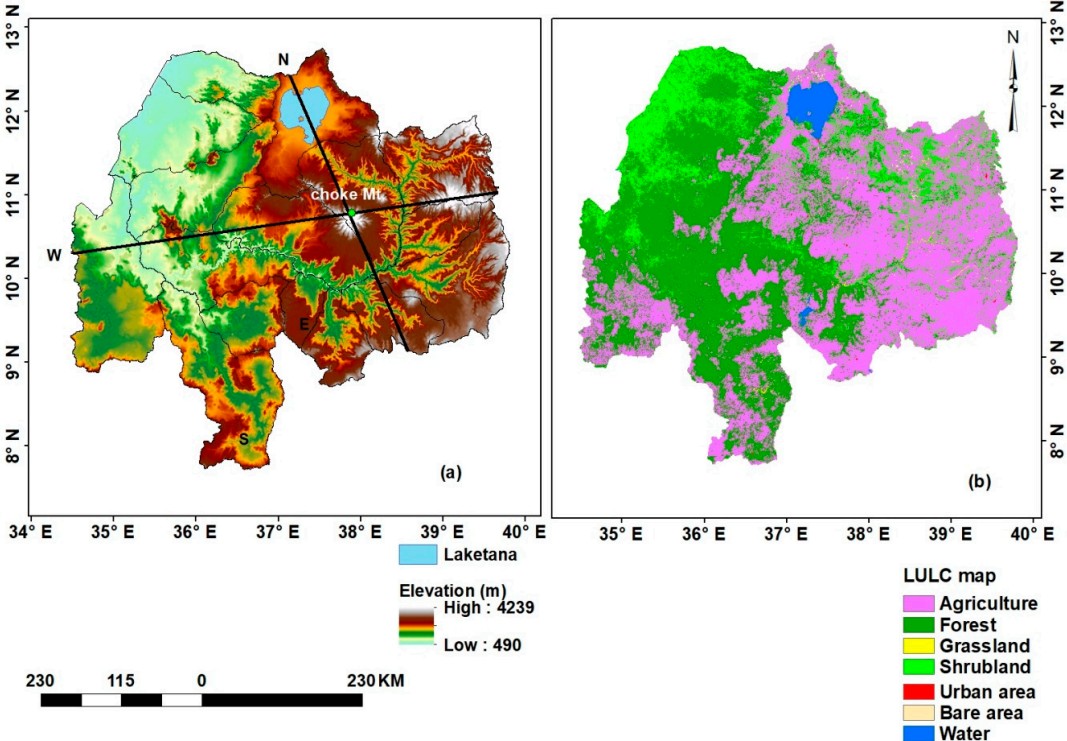

**Figure 2.** (**a**) Digital elevation model (DEM) with sub-basins of the Upper Blue Nile Basin (two transects from north to south-east (NS) and from west to east (WE)), and the green point shows the location of the peak "Choke" Mountain; (**b**) The land use land cover map of the UBNB for 2015 extracted from the European Space Agency's Climate Change Initiative (ESA CCI) land cover map.

In this study, the trend analysis was done using the Mann–Kendall (MK) trend test [67,68]. According to Mann, the null hypothesis $H_0$ states that a data series is serially independent and identically distributed with no monotonic trend. The alternative hypothesis $H_1$ is that the data follows a monotonic trend. In a two-sided test for a trend at a significance level of $\alpha$, $H_0$ should be rejected, and $H_1$ is accepted if $|Z| > z_{\alpha/2}$, where $F_N(z_{\alpha/2})$ is the standard normal cumulative distribution function and $Z$ is the test statistic used to identify the direction of the trend and its significance (Equation (3) to Equation (5)).

$$Z = \begin{cases} \frac{S-1}{\sqrt{var(s)}} & if\, S > 0 \\ 0 & if\, S = 0 \\ \frac{S+1}{\sqrt{var(s)}} & if\, S < 0 \end{cases} \tag{3}$$

where $S$ is the MK test statistic, which measures the trend in the data and is defined as:

$$S = \sum_{i=1}^{n-1} \sum_{j=i+1}^{n} sgn(x_j - x_i) \tag{4}$$

where $x$ is the sequential data values, and $n$ is the length of the dataset.

$$sgn(\theta) = \begin{cases} 1 & if\, \theta > 0 \\ 0 & if\, \theta = 0 \\ -1 & if\, \theta < 0 \end{cases} \tag{5}$$

Kendall indicated that the distribution of $S$ may be well approximated by a normal distribution with mean zero and variance (Equations (6) and (7)) under the assumption of no trend:

$$E(S) = 0 \tag{6}$$

$$var(S) = \left[ \frac{n(n-1)(2n+5) - \sum_{i=1}^{m} t_i\,(t_i-1)(2t_i+5)}{18} \right] \tag{7}$$

where $m$ is the number of tied groups and $t_i$ is the size of the $i$th tied group.

Since the MK test does not provide the magnitude of the trend, the Theil-Sen Slope Estimator [69] in Equation (8) has been used to estimate the magnitude of the trend ($\beta$).

$$\beta = Median = \left[ \frac{T_j - T_i}{j - i} \right], \forall i < j \tag{8}$$

where $\beta$ is the slope between data point $T_j$ and $T_i$ measured at times $j$ and $i$, respectively. In this paper, the trend analysis has been carried out using "ZYP" R package [70], developed based on Yue et al.'s [71] trend-free pre-whitening method. The trend result has been evaluated at the significance level of $\alpha = 0.05$. This implies that the null hypothesis is rejected when $|Z| > 1.96$ in Equation (3), thus, $Z > 1.96$ indicates a significant increasing trend and $Z < -1.96$ indicates a significant decreasing trend.

### 2.3.2. The Relationship Between Soil Moisture and Rainfall in Autumn

To understand the magnitude of the association between autumn soil moisture (ASM) and rainfall grid-level, a Pearson correlation was performed using the "corLocal" function in R.

The Dgital Elevation Model (DEM) and land cover map given in Figure 2a,b obtained from the Shuttle Radar Topography Mission (SRTM) and ESA CCI land cover, respectively, were used to gauge

the impact of topography and land cover classes on the correlation between ASM and rainfall in autumn. To investigate the effect of topography, first two transects that cross both incised river gorges and the pick mountains at the center of the basin that runs from the north to south-east (NS) and from west to east (WE) directions were developed (Figure 2a). Then, we assessed the correlation between ASM and annual rainfall along the two transects. The anomalies of rainfall and ASM were also correlated to analyze the effect of rainfall on ASM variability. Further, the effect of the relationship between rainfall and evapotranspiration to ASM change over the UBNB was investigated.

## 3. Results

### 3.1. Variability and Trends of Soil Moisture

Figure 3 presents the spatial pattern of multiyear (1992 to 2017) mean autumn (average of September to November) soil moisture and coefficient of variation (CV, %), while Figure 4 provides the relation between the CV and long-term mean ASM in the UBNB. Figure 5 depicts the annual soil moisture anomalies for the autumn season. In addition, Table 2 gives the spatial mean and CV of soil moisture for three major land cover classes of the UBNB over the three decades.

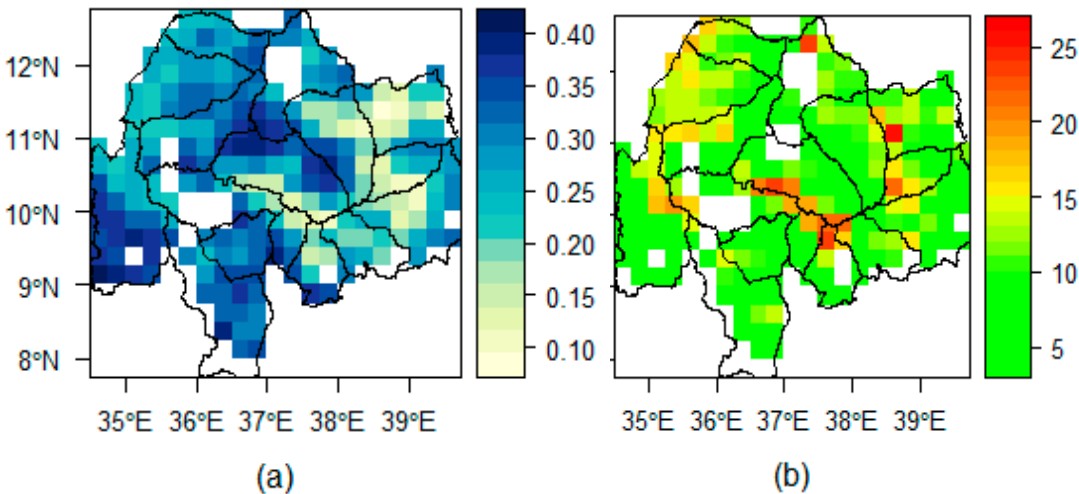

**Figure 3.** Spatial patterns of (**a**) mean autumn (September to November average) soil moisture (ASM) (m$^3$/m$^3$) and (**b**) coefficient of variation (CV, %) over the UBNB in Ethiopia for the period of 1992 to 2017. Note that the white regions (open pixels) show the pixels with missing values in the ESA CCI soil moisture dataset.

The multiyear mean soil moisture analysis showed a general overview of basin soil moisture distribution in the autumn season (Figure 3a). The result reveals that ASM in the basin varied considerably (0.09 to 0.38 m$^3$/m$^3$) with an average of 0.26 m$^3$/m$^3$ (Figure 3a). It is worth noting that the spatial patterns of mean ASM showed local characteristics and regions with remarkably high values are mainly located in the central, south-western and southern tips of the basin, while the lowest values majorly occur over the eastern parts (Figure 3a). On the other hand, the CV ranged from 2.8% to 28% (Figure 3b), which represents low-to-moderate variability over the study basin [72]. The relatively high variability of ASM generally occurred over lower mean soil moisture regions, while lower variability appeared over high soil moisture locations of the UBNB. To further investigate how changes in mean autumn soil moisture affect the CV over the basin, we plotted the relationship between mean soil moisture and CV for the period of 1992 to 2017 (Figure 4). The result indicated significant negative correlation (r = −0.6, $p < 0.01$) between mean soil moisture and soil moisture variability (expressed by the CV) over the UBNB, and the CV typically reduced with an increase in mean soil moisture.

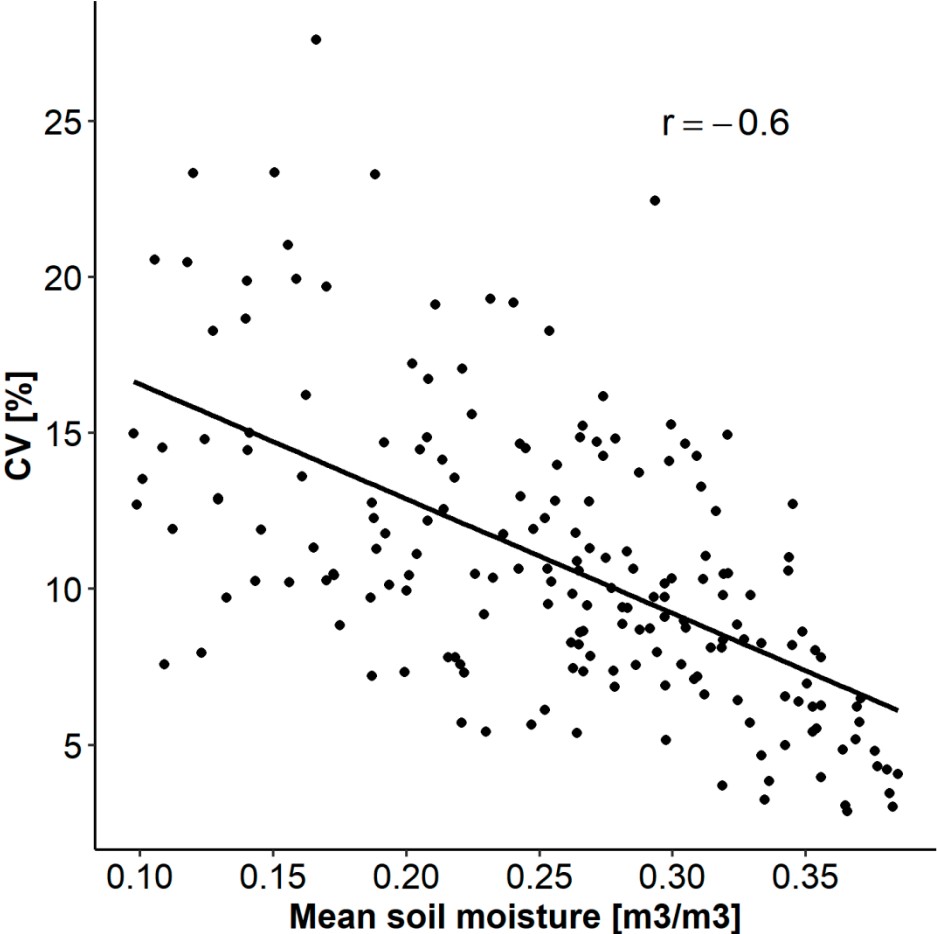

**Figure 4.** The relation between the coefficients of variation (CV, %) and long-term mean ASM in the UBNB.

In addition, Table 2 shows ASM variability for the dominant land cover classes of the basin arranged for three decades. Variations in the mean soil moisture content were found across the three land cover classes. Looking to the mean soil moisture values for all three decades, the shrub land was drier than agricultural and forest lands. As shown in Table 2, the analysis did not show much difference in the mean values of soil moisture over the agricultural and forest land covers. However, the ASM temporal variability exhibited a considerable variation across the different land cover classes of the basin. Accordingly, the highest variability was observed over forestland even for nearly similar or higher mean values to agricultural and shrub lands.

For example, during the second decade (2001 to 2010), agricultural, forest, and shrub lands resulted in a mean soil moisture of 0.24, 0.22, and 0.21 $m^3/m^3$ and CV of 11.20%, 27.07%, and 12.88%, and for the third decade (2010 to 2017), they resulted in a mean soil moisture of 0.27, 0.29, and 0.25 $m^3/m^3$ and CV of 3.90%, 8.28%, and 7.04%, respectively.

Annual soil moisture anomalies for the autumn season in Figure 5 showed that there are variations in the amount and distributions of ASM at different years and parts of the basin. Consequently, periods wetter than an average soil moisture year were mainly observed in 1997, 1998, 2016, and 2017, while the most critical periods that were drier than an average year were prominently noted in 1995, 1996, 2002, 2003, 2004, 2005, 2009, and 2015.

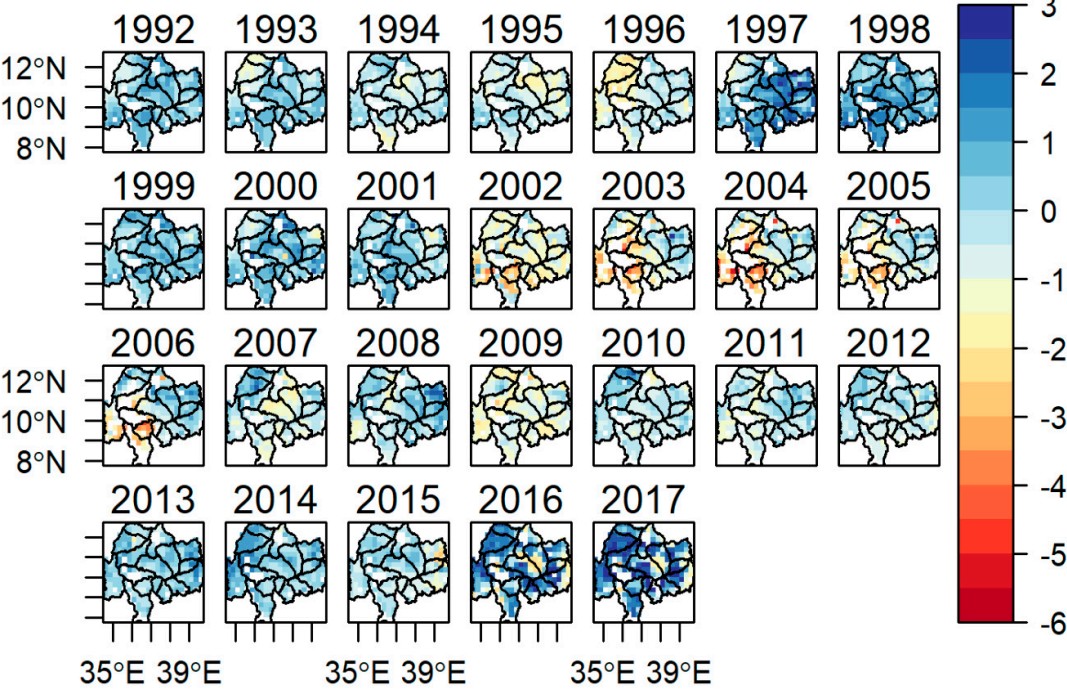

**Figure 5.** Standard soil moisture anomalies for autumn season in the UBNB indicating the magnitude of departure from long-term mean ASM over a period of 26 years of observation (1992 to 2017). A negative value represents periods drier than normal years, while a positive value indicates wetter than normal soil moisture periods.

**Table 2.** Decadal variability of autumn soil moisture (ASM) across the major land use land cover classes in the UBNB, Ethiopia. A lookup table for the land use land cover classification system is provided in Appendix B.

| LULC Type | ESA CCI LCM Code | 1992 to 2000 | | 2001 to 2010 | | 2010 to 2017 | |
|---|---|---|---|---|---|---|---|
| | | Mean | CV (%) | Mean | CV (%) | Mean | CV (%) |
| Agriculture | 10,11,12,20,30,40 | 0.28 | 5.27 | 0.24 | 11.20 | 0.27 | 3.90 |
| Forest | 50,60,61,62,70,72,80,81,82,90,100,160,170 | 0.28 | 5.01 | 0.22 | 27.07 | 0.29 | 8.28 |
| Shrub | 120,121,122 | 0.24 | 5.93 | 0.21 | 12.88 | 0.25 | 7.04 |

The ASM for the other years showed a small deviation (negative or positive anomaly) from the long-term mean, confirming that the basin has near-average autumn soil moisture during these periods.

In addition, the trend analysis was undertaken to explore the spatial consistency of ASM over the study periods from 1992 to 2017. The soil moisture monotonic trend and pixels having $p < 0.05$ values are depicted in Figure 6. The results of MK trend analysis for mean autumn (average of September to November) soil moisture showed different spatial-temporal trends in the UBNB. The trend ranges from $-0.0094$ to $0.0055$ m$^3$/m$^3$ per year for the autumn season, while an average over the entire basin indicated a wetting trend for the past 26 years (1992 to 2017) at a rate of $0.00024$ m$^3$/m$^3$ per year (Figure 6a). Considering the spatial coverage of the trends, both the drying and wetting trends proportionally cover the study area, with relatively high magnitudes for the wetting trends. The autumn time series reveals a significant wetting trend (ranging from $0.001$ to $0.006$ m$^3$/m$^3$ per year for the autumn season) that primarily occurred over the northwest region of the basin and covers about 18% of the total basin (Figure 6b). A significant drying trend has also been noted over the southeast of the UBNB and only covers about 4.5% of the total basin area (Figure 6b).

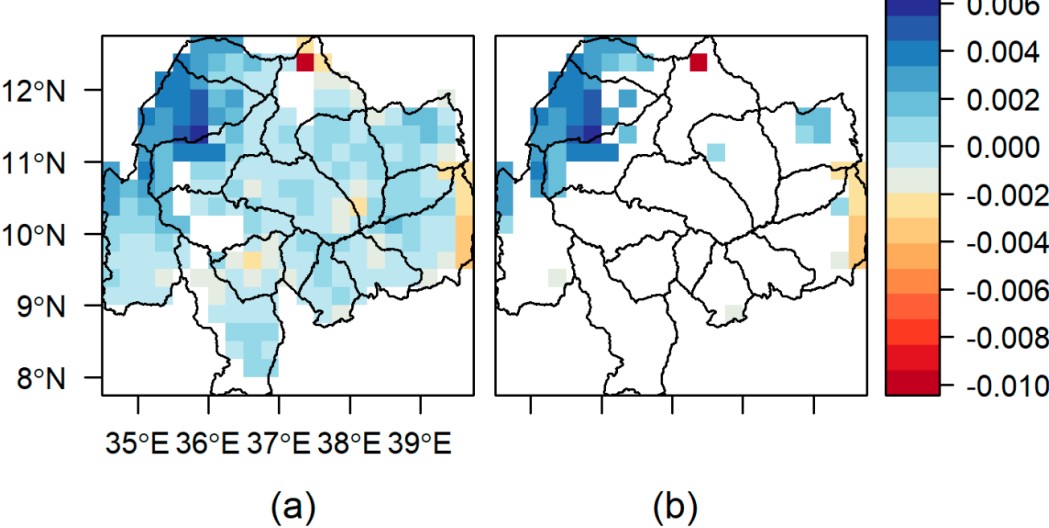

**Figure 6.** This figure shows (**a**) the monotonic trends ($m^3/m^3$ year$^{-1}$ for ASM) and (**b**) pixel having a significant ($p < 0.05$) increase or decrease trend of residual soil moisture for the period of 1992 to 2017. The positive and negative values indicate the wetting and drying trends of soil moisture in the UBNB, respectively.

### 3.2. Variability and Trends of Rainfall

Figures 7 and 8 present the long-term annual mean and anomalies of rainfall for the period from 1992 to 2017, respectively. The UBNB receives an annual rainfall > 2000 mm and its distribution has local characteristics, and the maxima and strong rainfall gradients are oriented along the central and southern tip of the basin (Figure 7). The lowest total annual rainfall occurred over the eastern and western margins of the basin. The wettest periods (e.g., 1996, 1998, 2000, 2006, 2008, and 2017) and driest periods (e.g., 1992, 1994, 1995, 2002, 2009, and 2015) of the UBNB were well captured in the analysis using the CHIRPS satellite rainfall product (Figure 8). Figures 9 and 10 provide the rainfall trends and masks of their respective pixel values, having *p* values <0.05 for the study period, respectively.

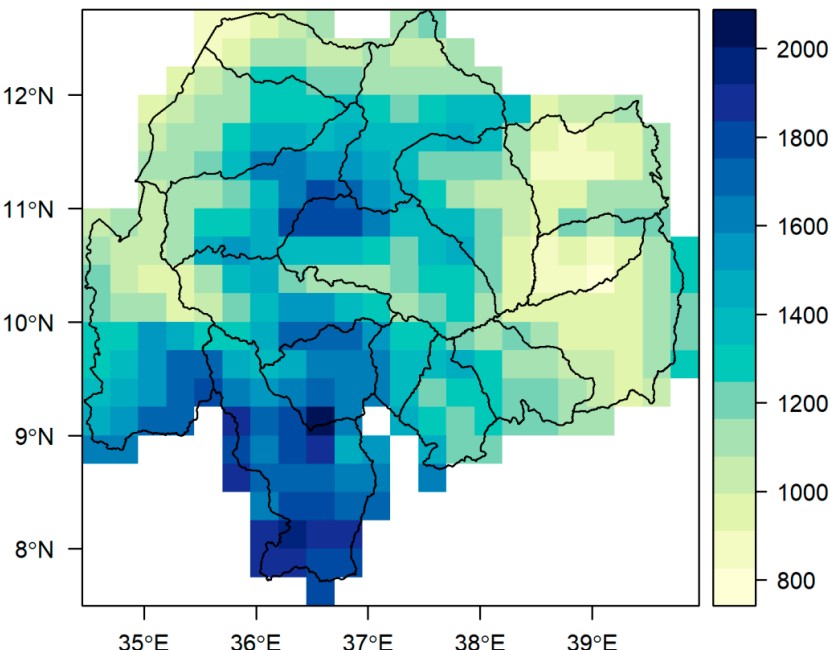

**Figure 7.** The spatial distribution of mean annual rainfall (mm) over the Upper Blue Nile Basin (UBNB) in Ethiopia (1992 to 2017).

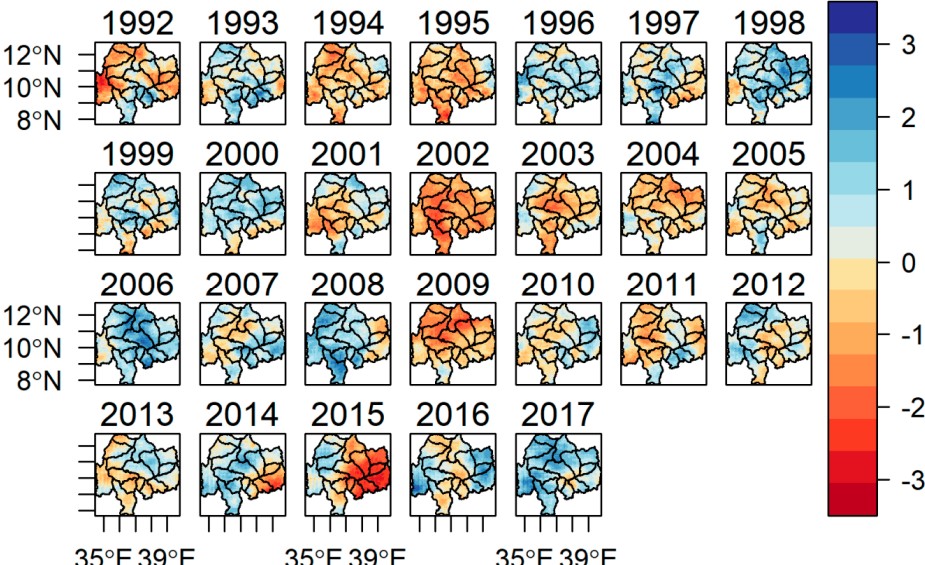

**Figure 8.** Standard rainfall anomalies for annual rainfall in the UBNB indicating the magnitude of departure from long-term mean rainfall over a period of 26 years observation (1992–2017). A negative value represents periods of below-normal rains (drought), while positive values indicate above-normal rains (with the possible risk of flood).

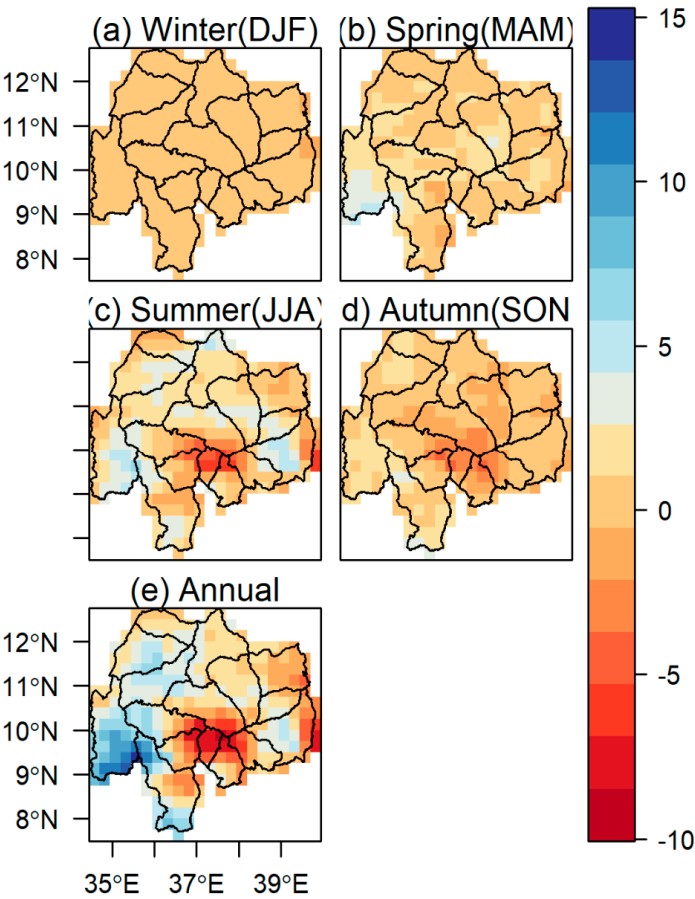

**Figure 9.** Monotonic trends for seasonal (mm/season) rainfall: (**a**) winter, (**b**) spring, (**c**) summer, (**d**) autumn, and (**e**) annually (mm/year) in the UBNB in Ethiopia for the period of 1992 to 2017. The annual rainfall is characterized by four distinct seasons: winter (December, January, February), spring (March, April, May), summer (June, July, August), and autumn (September, October, November).

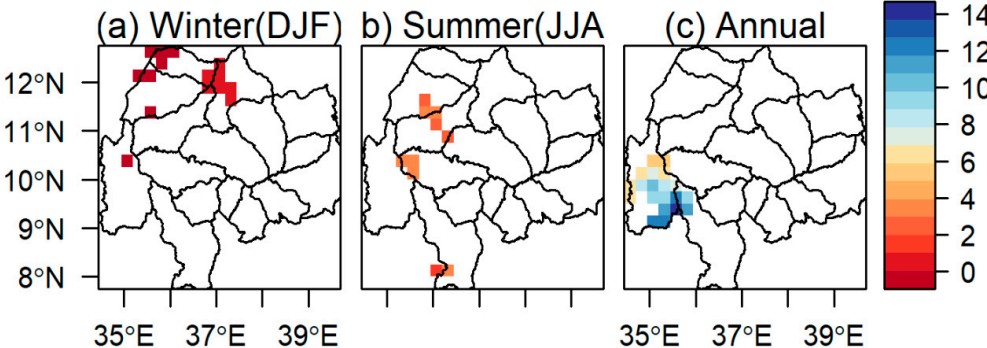

**Figure 10.** Zones with significant ($p < 0.05$) increasing or decreasing trends for (**a**) winter, (**b**) summer, and (**c**) annually in the UBNB in Ethiopia for the period of 1992 to 2017. No significant trends were found over the spring and autumn seasons.

When the seasonal rainfall trends were averaged over the entire basin, an increasing trend was observed in the spring, summer, and autumn rainfall with the rate of change 0.936 mm year$^{-1}$, 1.027 mm year$^{-1}$, and 0.071 mm year$^{-1}$, respectively (Figure 9b–d). However, the average rainfall trend during the winter season indicated a decreasing trend with the rate of change $-0.0014$ mm year$^{-1}$ (Figure 9a). Although it covers a small portion of the study basin, statistically significant increasing trends at $p < 0.05$ were identified during the winter (with the rate of change up to 0.313 mm year$^{-1}$) and summer (up to 3.714 mm year$^{-1}$) rainfall seasons (Figure 10a,b). The significant trends of rainfall during the winter and summer season are majorly marked in the north and west of the UBNB, respectively (Figure 10a,b). Figure 10 also revealed that there was no significant trend in spring and autumn rainfall, while the annual rainfall trend average over the entire basin showed an increasing trend with the rate of change 1.819 mm year$^{-1}$ (Figure 9e), with a significant increasing trend reaching up to 13.714 mm year$^{-1}$ located in the southwest of the basin (Figure 10c).

### 3.3. Correlation Coefficient Between Autumn Soil Moisture and Rainfall

Figure 11 illustrates the pixel-level correlation computed between soil moisture for the autumn season and rainfall over the UBNB from 1992 to 2017. The correlation coefficient (r) indicates how much ASM can be explained by rainfall in the UBNB. In general, the soil moisture for the autumn season exhibits a positive correlation with spring, autumn, and annual rainfall (Figure 11b,d,e), and a negative correlation with winter and summer rainfall over most parts of the UBNB (Figure 11a,c).

Notably, one may expect an increase in ASM due to the predecessor wet summer rainfall in comparison to the relatively low rainfall during the spring and autumn seasons. However, this may not sometimes be proportional to the amount of rainfall received because of the surface runoff, intense evapotranspiration, etc., which usually happens during the wet periods [73]. The ASM conferred a strong positive correlation (r > 0.5) with the spring, autumn, and annual rainfall, which covers 6%, 35%, and 15% of the total basin area, respectively. Distinctly, the correlation between ASM and wet summer rainfall was negative over a significant portion of the basin, although less than 3% of the total basin area showed a statistically strong correlation (r < −0.5). The ASM and winter rainfall have also shown both a negative and positive correlation, with a significant correlation (r > 0.4) over a considerable portion of the basin. Overall, the correlation analysis indicates that ASM was strongly correlated to rainfall during the autumn and spring seasons than that of the summer season.

Generally, the correlation between ASM and rainfall reduced from the west to east region (or from low- to high-elevation areas) of the basin (Figure 11). Since the high variation of terrain elevation characterizes the UBNB, the spatial correlation between soil moisture and rainfall could be affected by a change in topography. Consequently, the study further assessed the effect of elevation variation on the correlation between autumn soil moisture and annual rainfall. Two transects that run from the north to south-east (NS) and from west to east (WE) directions were developed (Figure 2a). Then,

the correlation between ASM and annual rainfall was assessed along the two transects (Figure 12). The result from both NE (Figure 12a) and WE (Figure 12b) indicates that the correlation between ASM and annual rainfall is affected by the variation in elevation as the correlation reduces as the elevation increases. The lowest correlation between soil moisture and rainfall has occurred over peak areas of the basin.

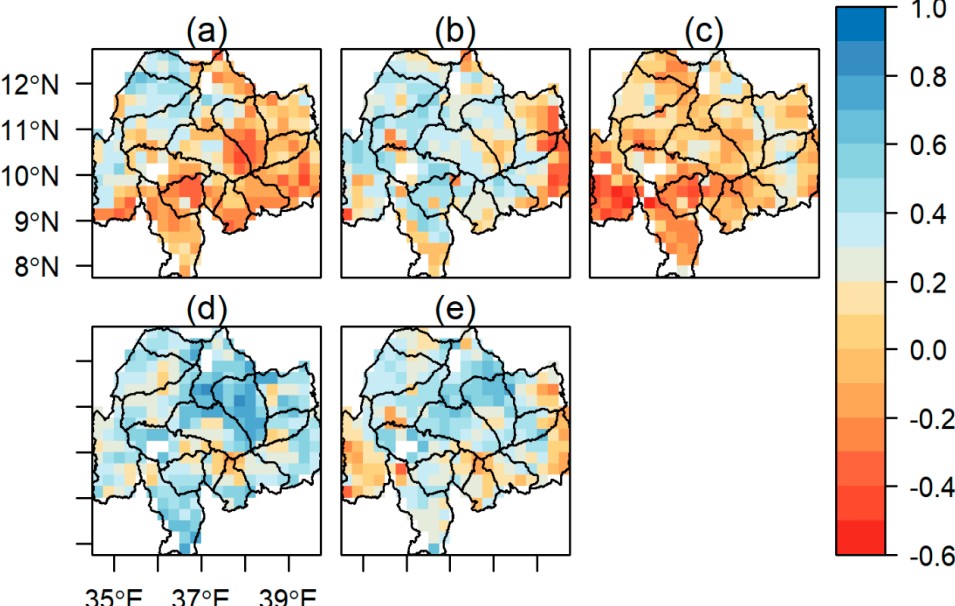

**Figure 11.** Pearson correlation (r ranging from −1 to 1) between ASM and rainfall for (**a**) winter, (**b**) spring, (**c**) summer, (**d**) autumn season, and (**e**) annually (1992 to 2017). The correlation coefficient (r ≥ |0.39|) is significant at $p < 0.05$.

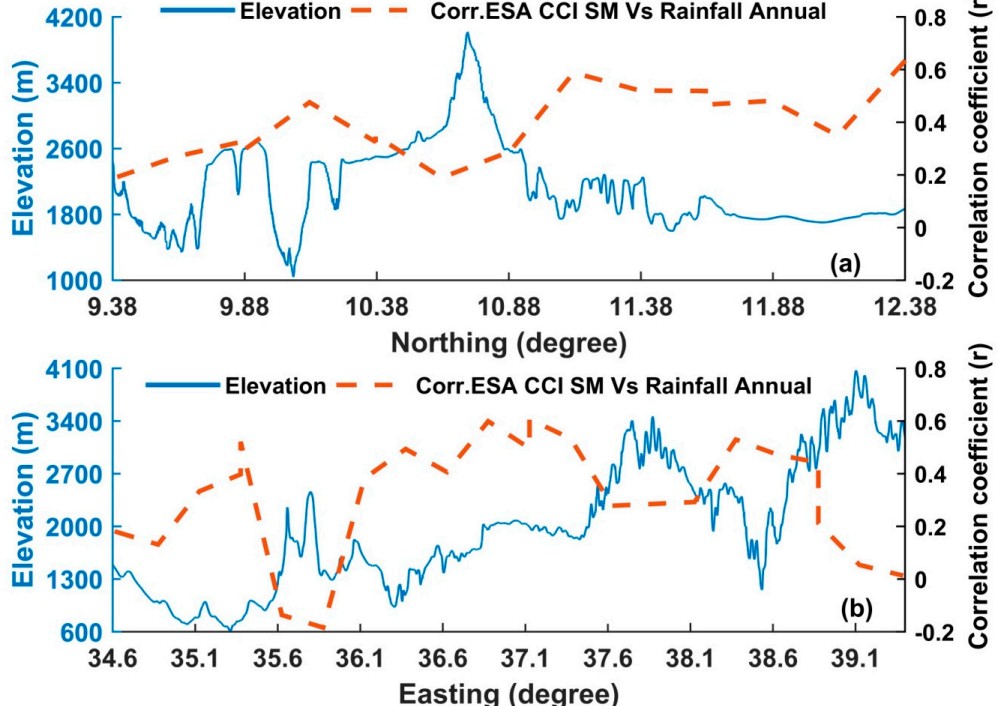

**Figure 12.** The correlation between autumn soil moisture (ASM) and annual rainfall along transects from (**a**) from north to south-east and (**b**) from west to east (WE) (1992 to 2017).

The degrees to which land cover types affect the correlation between ASM and rainfall could vary with vegetation type, density, and seasons. Table 3 shows the results of the spatial correlation between ASM and rainfall over the dominant land cover classes of the basin. The correlation between ASM and rainfall respond distinctively across the different seasons and land cover classes. For example, the highest correlation between ASM and spring rainfall was observed over the forestland (r = 0.51); however, autumn (r = 0.19) and summer (r = −0.28) rainfall showed their lowest positive and highest negative correlation in forestland, respectively.

**Table 3.** The correlation between ASM and rainfall over the major land use land cover classes of the UBNB.

| LULC Type | ESA CCI LCM Code | Rainfall | | | | |
|---|---|---|---|---|---|---|
| | | Winter | Spring | Summer | Autumn | Annual |
| Agriculture | 10,11,12,20,30,40 | −0.16 | 0.37 | −0.04 | 0.40 * | 0.35 |
| Forest | 50,60,61,62,70,72,80,81,82,90,100,160,170 | −0.12 | 0.51 * | −0.28 | 0.19 | 0.26 |
| Shrub | 120,121,122 | −0.10 | 0.46 * | 0.12 | 0.40 * | 0.46 * |

\* significant at $p < 0.05$.

The highest correlation for the annual (r = 0.46) and positive correlation for summer (r = 0.12) rainfall has been indicated over the shrub land. The ASM and rainfall in autumn have shown the same correlation values for both shrub land and agricultural land. Over the agricultural field, the correlation between soil moisture and rainfall resulted in an optimal correlation in the autumn (r = 0.40), annual (r = 0.37), and spring (r = 0.37) seasons. The correlation in winter was negative for all three land cover classes with a better correlation over the agricultural land. However, the correlation between summer rainfall and ASM over the agricultural field was insignificant.

*3.4. Impact of Rainfall Variability on Soil Moisture Dynamics*

To examine how the variability of rainfall affects ASM in the UBNB, the correlation between ASM and rainfall anomalies was analyzed. Their correlation values at the seasonal and annual scale were averaged over the entire basin, as shown in Table 4. Moreover, Figures 13 and 14 show the temporal (time series) variation and the strength and sign of the correlation between ASM and rainfall anomalies.

**Table 4.** The correlation between ASM anomalies at different depths and rainfall anomalies at the seasonal and annual scale from 1992–2017.

| Soil Moisture | Rainfall | | | | |
|---|---|---|---|---|---|
| | Winter | Spring | Summer | Autumn | Annual |
| ESA CCI SM | 0.12 | 0.46 * | 0.10 | 0.64 * | 0.64 * |
| FLDAS NOAH (0–10 cm) | −0.07 | 0.36 | 0.07 | 0.91 * | 0.69 * |
| FLDAS NOAH (0–40 cm) | −0.08 | 0.35 | 0.10 | 0.91 * | 0.71 * |
| FLDAS NOAH (0–100 cm) | −0.08 | 0.35 | 0.12 | 0.91 * | 0.73 * |

\* significant at $p < 0.05$.

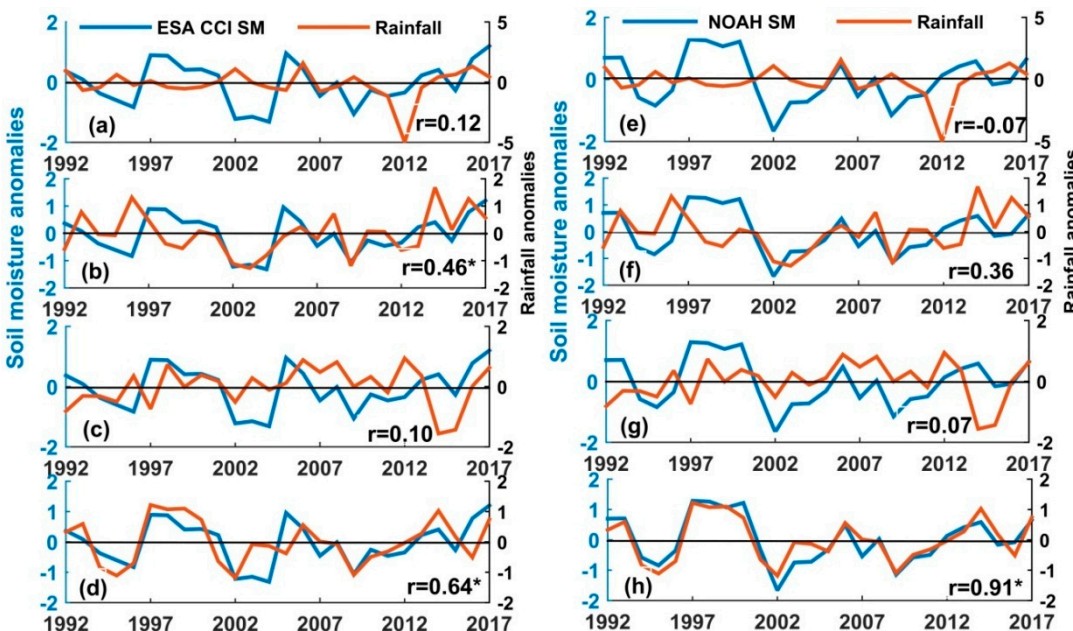

**Figure 13.** Time series of the autumn soil moisture (ASM) anomalies (estimated using ESA CCI) and rainfall anomalies in the preceding (**a**) winter: December, January, and February (DJF), (**b**) spring: March, April, and May (MAM), (**c**) summer: June, July, and August (JJA), and (**d**) autumn: September, October, and November (SON) averaged over the entire basin. Figure (**e**–**h**) are the same as Figure (**a**–**d**), but ASM anomalies were calculated using FLDAS NOAH at 0 to 10 cm depth. The correlation coefficients (r) between two datasets are also shown; the asterisk represents the values significant at $p < 0.05$.

The use of temporal anomalies could reduce the impacts of static properties (i.e., topography and soil properties) on soil moisture dynamics and thus are assumed to primarily reflect the impact of climate variables [74]. The correlation analysis was undertaken using soil moisture derived from ESA CCI, which denotes the top few centimeters of the soil, and FLDAS NOAH soil moisture that represents soil moisture at 0–10, 0–40, and 0–100 cm depths. A weighted average method was used to calculate soil moisture at 0–40 and 0–100 cm depths. The correlation between ASM and rainfall in the autumn or the preceding three seasons was positive (Figure 13a–d, Table 4), although the contribution of the wet summer rainfall to ASM was insignificant (relatively low) in the UBNB. On the other hand, the correlation between ASM and annual rainfall anomalies was significant and revealed that the total amount of rainfall in each year has a considerable impact on the change in the ASM (Figure 14a, Table 4). The same result has been observed for FLDAS soil moisture at different depths (Figure 13e–h, Table 4), except for winter rainfall, which was negatively correlated with ASM. The ASM derived from FLDAS NOAH showed that the contributions of spring rainfall decreased, while the contribution of the current rainfall in autumn was substantial, and its correlation with the ASM was robust (r = 0.91) at all depths of the soil. In general, the result indicated the significant contribution of rainfall to ASM at different depths, although the magnitude of contribution was large for sub-surface soil moisture.

In addition, the wetting trends of ASM derived from ESA CCI were mainly associated with the increasing trend for spring, summer and autumn rainfall, which was generally exhibited in the northwest and southwest of the basin (Figures 6 and 9). The spatial similarity in the trends of ASM with autumn rainfall was generally higher than the other temporal scales, although some noticeable differences existed between trends of rainfall and ASM. For instance, ASM showed a statistically significant increasing trend that was observed over the northwest of the basin where an increase in rainfall trend was not significant.

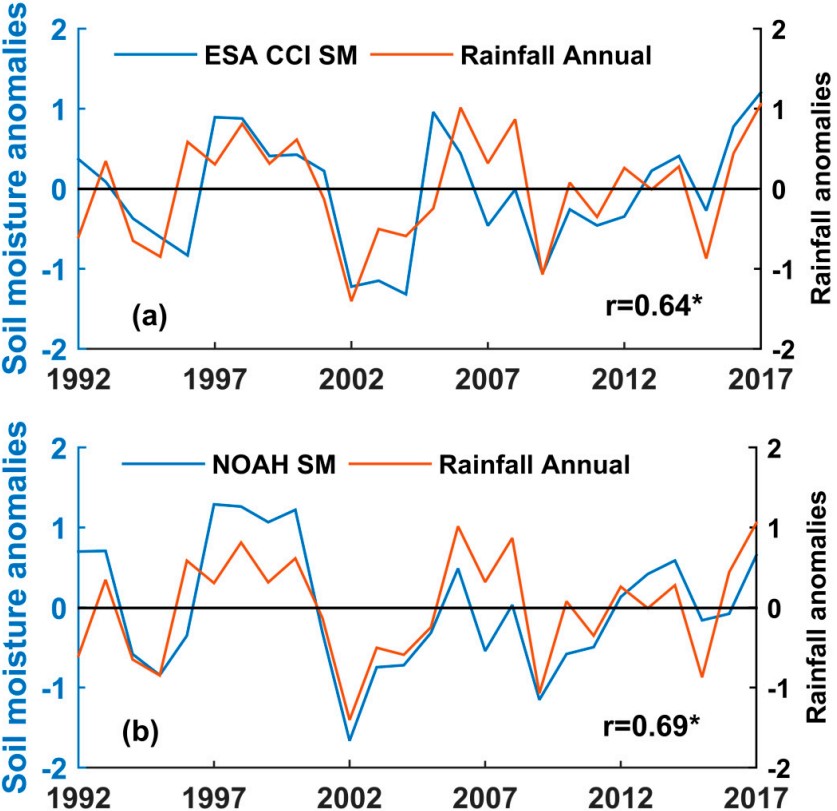

**Figure 14.** Time series of annual rainfall anomalies and ASM anomalies estimated using (**a**) ESA CCI and (**b**) FLDAS NOAH at 0 to 10cm depth averaged over the entire UBNB. The correlation coefficient (r) between the two datasets is also shown; the asterisk represents the value significant at *p* < 0.05.

### 3.5. The Effect of the Relationship Between Rainfall and ET on Soil Moisture

Besides rainfall, evapotranspiration (ET), which is the sum of soil evaporation and vegetation transpiration, is a crucial climate variable influencing the distribution and availability of soil moisture [75]. It is essential to determine the effect of the quantitative relationship between ET and rainfall on the temporal patterns of ASM in the UBNB. The ratio of ET to rainfall (RF) was calculated to determine their effect on the wetness and dryness of ASM derived from ESA CCI in the UBNB (Figures 15 and 16).

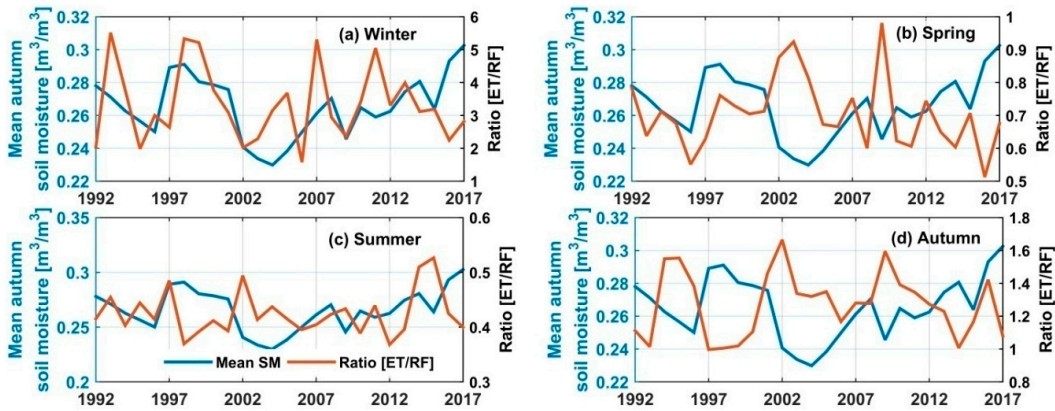

**Figure 15.** The effects of (**a**) winter, (**b**) spring, (**c**) summer, and (**d**) autumn rainfall (RF) and evapotranspiration (ET) (expressed as ET/RF ratio) on the temporal distribution of ASM in the UBNB from 1992 to 2017.

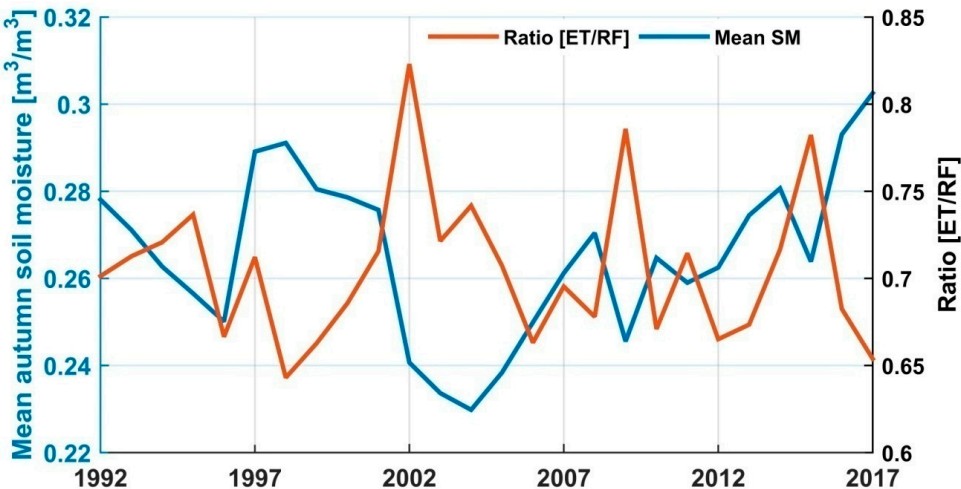

**Figure 16.** The effects of annual rainfall (RF) and evapotranspiration (ET) (expressed as ET/RF ratio) on the temporal distribution of autumn soil moisture over the UBNB from 1992 to 2017. We can see the highest ET/RF values over the known drought periods of 2002, 2009, and 2015 in the UBNB, as well as in Ethiopia in general [76].

In general, from Figures 15 and 16 we can see that the amount of mean soil moisture decreases with an increase in ET/RF ratio. Over the winter season (Figure 15a), ET has an average rate of 83.98 mm year$^{-1}$ and stays low since water availability is reduced due to the absence or significantly low rainfall (an average of 27.56 mm year-1) events in winter. Consequently, ASM has a positive correlation (r = 0.22) with ET/RF ratio. The significant correlation between ASM and winter ET (r = 0.41, $p < 0.05$) to ET/RF ratio indicates that ET explains the soil moisture variability better than rainfall in winter. The winter rainfall has a very low correlation (r = −0.13) with ASM as expected. In spring (Figure 15b), the rainfall (with an average of 219.70 mm year$^{-1}$) and ET (mean rate of 150.01 mm year$^{-1}$) have a strong positive correlation (r = 0.85) and ET increases with rainfall. Both the rainfall and ET have a significant positive correlation with ASM, with r = 0.49 ($p < 0.05$) and r = 0.46 ($p < 0.05$), respectively. However, rainfall increases more than ET; thus, ASM has a negative correlation (r = −0.47) with ET/RF ratio over the spring season. In summer (Figure 15c), rainfall exceeds (mean rainfall 764.97 mm year$^{-1}$) ET (average rate of 323.09 mm year$^{-1}$), but ASM has a small negative correlation with ET/RF ratio (r = −0.11). The summer rainfall has no correlation with autumn soil moisture (r = 0.016), and no apparent contribution to the amount and variability of ASM was observed during the summer season. Instead, summer ET showed a negative correlation (r = −0.24) with ASM over the UBNB. In autumn (Figure 15d), ET (with the mean rate of 351.09 mm year$^{-1}$) exceeds that of rainfall (an average of 282.15 mm year$^{-1}$) and ASM has a strong negative correlation (r = −0.57, $p < 0.05$) with the ET/RF ratio. However, the ASM showed a strong positive correlation (r = 0.56, $p < 0.05$) with autumn rainfall compared to the low negative correlation (r =−0.27) with autumn ET. Over the annual analysis (Figure 16), although both the mean annual rainfall (1294.63 mm year$^{-1}$) and mean ET (909.73 mm year$^{-1}$) have a positive correlation with autumn soil moisture, r = 0.57($p < 0.05$) and r = 0.24, respectively, the rainfall exceeds that of ET, and ASM has a significant negative correlation (r = −0.59, $p < 0.05$) with the annual ET/RF ratio.

## 4. Discussion

Soil moisture is a vital component of agriculture and hydrology. Its spatial-temporal change can considerably affect the socioeconomics of agrarian countries such as Ethiopia. Understanding the soil moisture dynamics and its associated response to climate variables (such as rainfall) is vital for proper water resource planning. This study investigated the spatial and temporal characteristics of residual soil moisture in autumn (September to November) over the UBNB in Ethiopia. The result indicated that the variability of soil moisture is affected by the extent of moisture available in the soil and found

that dry soil regions vary more than that of wet soil regions (Figure 4). Indeed, this result is compatible with the findings of different scholars who found an increase in soil moisture spatial variability with decreasing mean soil moisture values [77,78]. Similarly, a comparable result for rainfall variability and mean rainfall over the UBNB has been found (Figure A2, Appendix A), which indicates the direct impact of rainfall on the soil water content of the basin.

Further, the analysis showed that the moisture status of the soil was dependent on the land cover classes of the UBNB (Table 2). The different land cover classes of the basin could determine the distribution and availability of soil moisture through controlling the soil infiltration process and the rate of evapotranspiration [78]. In the study, the shrub lands show a drier condition than agricultural and forest land covers. Different scholars e.g., [78,79] found a similar result over shrub lands in comparison to other land cover classes. The drier soil moisture conditions beneath the shrub land could be associated with the higher density of plant roots for the shrubs than the other land cover classes, which eventually leads to a more significant loss of soil water through transpiration [78]. The agricultural and forest land covers of the basin showed a relatively wetter soil condition. Similarly, Feng et al. [79] have reported wetter soil conditions for cultivated lands. The tremendous infiltration rate of cultivated soil just after rainfall events and little soil water loss via transpiration from crops with lower leaf area index could contribute to the relatively wetter soil conditions over the agricultural fields. The wetter condition over the forest land could partly be explained by the maximum porosity of the soil in forest areas, less direct soil water evaporation under a higher coverage of plants, and little loss of water through transpiration at the surface soil for deep-rooted forests [28]. The relatively high soil moisture variability over forest areas could be explained by the diversity of trees within the forest, which may result in low soil moisture values around the trees and higher values in the interspaces between trees [28]. The high soil moisture variation over forest lands could also be due to the reduced quality and uncertainty of the ESA CCI soil moisture product over vegetated areas [80]. The annual anomalies for ASM are broadly comparable with wet and drought years of the basin identified by different scholars [76] and anomalously for dry and wet rainfall periods of the UBNB (Figure 8). These results suggest that the broad patterns exhibited by ESA CCI soil moisture data are likely reliable. The result could further indicate that the spatial and temporal distribution of rainfall is one of the significant factors which governs the variability of ASM, among the other climate variables.

Further, it is noted that the inter-annual variability of rainfall and ASM are very much associated. The annual soil moisture for the autumn season indicates a positive correlation with spring, autumn, and annual rainfall (Figure 11). Again, the changes in ASM anomalies at different soil depths also correspond to changes in the variability of rainfall in autumn and the previous spring seasons and showed a significant correlation (r = 0.35 to 0.91) (Figure 13, Table 4). The same result was also observed using the annual rainfall presented in Figure 14 and Table 4. The result implies that the impact of the current (autumn) and previous spring rainfall on ASM is positive and anomalies in ASM mainly originate from the variability of rainfall during the previous spring and autumn season as well as to the annual total. Although the intensity of rainfall is low in spring compared to autumn and the summer seasons, the spring rainfall could infiltrate into the soil layer and thus has a significant contribution to the amount of soil water left in the autumn season. Certainly, rainfall from the autumn season is the dominant contributor to the annual variability of ASM over the UBNB. Cai et al. [40] reported that rainfall and soil moisture have a high degree of correlation in autumn over eastern China. Likewise, Longobardi [81] indicated that the volume of rainfall occurs at the end of the wet season perhaps determines the amount and distributions of soil moisture at the beginning of the dry season. On the other hand, the contribution of summer rainfall to ASM variability is very insignificant (Figures 11 and 13, Table 4) and generally has a weak correlation with ASM compared to spring and autumn rainfall over the UBNB. This could be due to the increased rate of soil moisture depletion, reduced infiltration rate [82], and loss of incoming rain through surface runoff [73] over the summer growing season. According to Yang et al. [83], rainfall storage occurred in April and May (months in spring) because of soil water consumed over the wet months from June to mid-September

(summer season) and recovery of soil water from late September to October (months of the autumn season). Consequently, the contribution of summer rainfall to soil moisture storage is limited; therefore, this could partly be the reason for the low correlation between ASM and summer rainfall in the UBNB.

The findings of this study indicated that soil moisture response to rainfall variability is considerably controlled by the variation in topography and land use types of the UBNB (Figure 12, Table 3). The correlation between ASM and rainfall decreases with an increase in elevation because locations at a relatively higher elevation could have less available soil moisture due to gravity and exposure to sunlight warming, which may result in water drains downhill and higher evaporation rates, respectively [78]. Similar findings have been reported by Crave and Gascuel-Odoux [84]. Furthermore, the difference in the correlation between rainfall and ASM at different land-use types and seasons could be explained by the soil moisture build-up process over various land cover classes. In the shrub and agricultural lands, the build-up of soil moisture is relatively fast, and soil moisture peak is attained rapidly in comparison to the slow rate build-up process in forest land [85].

Figures 15 and 16, in general, imply that a significant portion of rainfall received by the UBNB returns to the atmosphere via evapotranspiration (ET). Accordingly, the amount of ASM over the periods of a higher ET/RF ratio in general reduced, while the magnitude of mean soil moisture peak corresponds to low ET/RF ratio (Figures 15 and 16). In each season the effects of rainfall and ET on ASM are varied. The highest ET/RF ratio in winter shows the dominance of ET over rainfall due to the absence or low amount of rainfall over this season. The result implies that in winter, ET could explain a considerable share of autumn soil moisture variance than that of rainfall. In spring, rainfall increases more than ET and produces a slightly higher rainfall than ET demands, suggesting that both the previous spring rainfall and ET have considerable contributions to the variability of autumn soil moisture in the UBNB, with the highest magnitude from spring rainfall. Over the wet summer season, despite rainfall exceeding that of ET, its contribution to ASM is very insignificant. Again, in autumn ET slightly exceeds rainfall and ASM has a strong negative correlation ($r = -0.57$, $p < 0.05$) with ET/RF ratio, which indicates that ET could explain the majority of autumn soil moisture variability in comparison to rainfall in autumn. However, the ASM has a strong positive correlation ($r = 0.56$, $p < 0.05$) with autumn rainfall compared to the low negative correlation ($r = -0.27$) with autumn ET, which may indicate the positive contributions of autumn rainfall to soil moisture rather than the loss of soil water via ET. This is because during autumn, crops and other vegetation canopies shade more and more of the ground area and may lead to less evaporation of water from the soil and thus a considerable portion of ET could come from the root zone via plant transpiration. Although other climate variables, including surface solar radiation, relative humidity, and wind speed might affect the inter-annual variability of residual soil moisture, we assume these effects to be small.

## 5. Conclusions

Understanding the availability and dynamics of residual soil moisture over the rainfed agricultural system, characterized by low crop production, is imperative for supplementary food and feed production in the off-season. In this study, we applied ESA CCI soil moisture products from 1992 to 2017 to assess the long-term trend and dynamics of residual soil moisture in the autumn (September to November) season and its linked response to the long-term variability of rainfall in the UBNB of Ethiopia. Besides, the mutual effect of rainfall and evapotranspiration on the variability of autumn soil moisture was analyzed. The basin was found to have soil moisture ranging from 0.09–0.38 $m^3/m^3$, with an average of 0.26 $m^3/m^3$ in autumn. The ASM time series resulted in the CV ranging from 2.8–28% and was classified with a low-to-medium variability. Moreover, ASM variability showed a strong relationship with the mean ASM, and the highest inter-annual variability occurred over low mean ASM areas of the basin. The mean and variability of ASM changes with the different land-cover classes of the basin. In general, the MK monotonic trend analysis for ASM revealed that the UBNB had experienced a wetting trend for the past 26 years (1992–2017) at a rate of 0.00024 $m^3/m^3$ per year. Besides, the study provided strong evidence that the previous spring and current autumn rainfall could

explain a considerable portion of ASM in the basin. Furthermore, the result indicates that the rainfall degree of influence on the characteristics of ASM could be induced by topography and dominant land cover classes of the study basin. In addition, the behavior of the mean ASM could be determined by the effect of soil water loss through evapotranspiration.

To conclude, the ESA CCI soil moisture product provides valuable insights into the spatial and temporal characteristics of autumn soil moisture over the UBNB. Thus, ESA CCI soil moisture estimates could be used as an alternative data to monitor the extent and dynamics of soil moisture over the data scarce regions such as the UBNB. However, high-resolution soil moisture datasets are still crucial for better understanding the characteristics of soil moisture in a complex topographic area of the UBNB. The information provided in this study could provide pertinent information to comprehend the soil moisture status of the basin in the off-season and its potential to support additional short or medium cycle cropping. Further, the link between soil moisture and rainfall presented in this study could play an important role to predict the extent and conditions of residual soil moisture in advance. Future studies should include the contributions of land use and land cover change in the extent and dynamics of soil moisture in the basin.

**Author Contributions:** G.A., T.T. and B.G. envisioned and designed the research; G.A. performed the data collection; G.A., T.T. and B.G. analyzed the results; G.A. wrote the original manuscript and T.T., and B.G. reviewed the manuscript. All authors have read and agreed to the published version of the manuscript.

**Funding:** This research was funded by Geospatial Data and Technology Center of Bahir Dar University (Grant No. BDU/RCS/GDTC/2009-04) and Entoto Observatory and Research Center postgraduate research fund.

**Acknowledgments:** The authors would like to thank the European Space Agency (ESA), NASA, and USGS for providing satellite and model products.

**Conflicts of Interest:** The authors declare no conflict of interest.

**Appendix A**

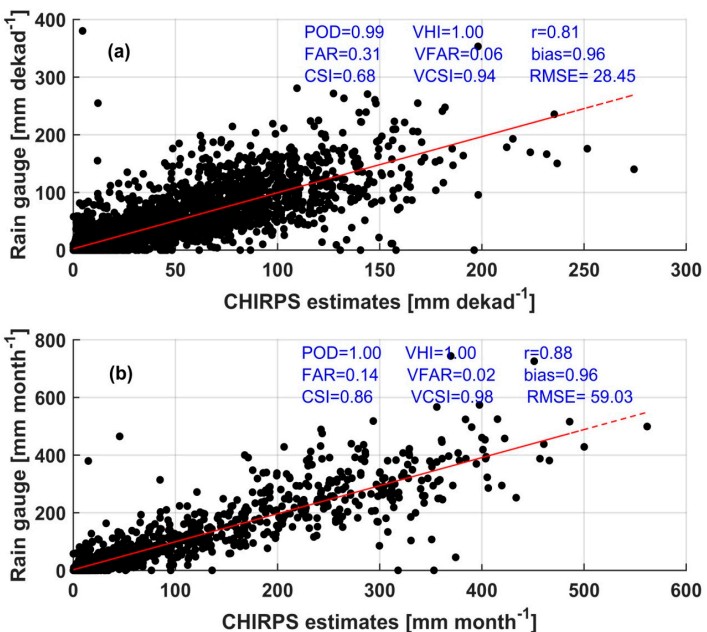

**Figure A1.** Scatter plot between rain gauge observations and CHIRPS rainfall estimates at dekadal (**a**) and monthly (**b**) temporal scale over the Upper Blue Nile basin for the period of 2000–2015. Probability of detection (POD), false alarm ratio (FAR), critical success index (CSI), volumetric hit index (VHI), volumetric false alarm ratio (VFAR), volumetric critical success index (VCSI), correlation coefficient (r), bias, and the root mean square error (RMSE). The perfect score for POD, CSI, VHI, VCSI and bias is 1, while 0 is the perfect score for FAR and VFAR. The RMSE values are presented in millimeters (mm) (See Ayehu et al., 2018 for the details of the analysis).

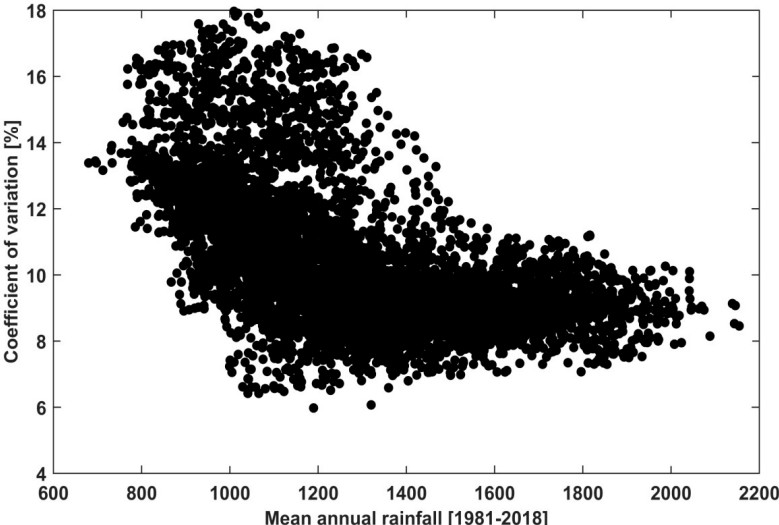

**Figure A2.** The relationship between the coefficients of variation (%) and long-term mean annual rainfall (mm). The CV increases with a decrease in mean annual rainfall with the coefficient of correlation (r) equal to −0.57.

## Appendix B

**Table A1.** A look up table for reclassification of land cover types (adapted from Land Cover CCI: product user guide, Version 2).

| Classes Considered in this Study | Land Types used in the CCI–LC Maps | |
|---|---|---|
| Agriculture | 10,11,12 | Rainfed cropland |
| | 20 | Irrigated cropland |
| | 30 | Mosaic cropland (> 50%) / natural vegetation (tree, shrub, herbaceous cover) (< 50%) |
| Forest | 40 | Mosaic natural vegetation (tree, shrub, herbaceous cover) (> 50%) / cropland (< 50%) |
| | 50 | Tree cover, broadleaved, evergreen, closed to open (> 15%) |
| | 60,61,62 | Tree cover, broadleaved, deciduous, closed to open (> 15%) |
| | 70,71,72 | Tree cover, needleleaved, evergreen, closed to open (> 15%) |
| | 90 | Tree cover, mixed leaf type (broadleaved and needleleaved) |
| | 100 | Mosaic tree and shrub (> 50%) / herbaceous cover (< 50%) |
| | 160 | Tree cover, flooded, fresh or brakish water |
| | 170 | Tree cover, flooded, saline water |
| Grassland | 110 | Mosaic herbaceous cover (> 50%) / tree and shrub (< 50%) |
| | 130 | Grassland |
| Wetland | 180 | Shrub or herbaceous cover, flooded, fresh-saline or brakish water |
| Urban | 190 | Urban |
| Shrubland | 120, 121, 122 | Shrubland |
| Bare area | 200, 201, 202 | Bare areas |
| Water | 210 | Water |

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
