# Peer review of "Monitoring Residual Soil Moisture and Its Association to the Long-Term Variability of Rainfall over the Upper Blue Nile Basin in Ethiopia"

_remotesensing, doi:10.3390/rs12132138_

Round 1

Reviewer 1 Report

Residual soil moisture is a very important hydrological parameter, which has very important significance for human life and agricultural production. The article is no problem in terms of structure, which is logical and legible. It summarized the work of a lot of previous studies in detail and draws some interesting conclusions. The specific issues are as follows:

(1) The quality of the graphics is poor, it is recommended to improve the quality.

(2) The research area of the article is too small, and the results and rules revealed may have obvious contingency and limitation. Why don't the authors choose a larger area for research?

(3) Many of the discussion parts are useless or repetitive content, such as lines 536-543, which do not provide any useful information. These contents have been mentioned in the summary and conclusion.

(4) The conclusion part is too long-winded. It is recommended to simplify the conclusion part. A lot of information does not need to be repeated again and again.

(5) Figure. 15 is invisible.

Reviewer 2 Report

The manuscript explored ways to utilize a global soil moisture product to understand soil moisture availability to improve farmer production.  I think that the premise of the paper was good. 

There is no mention of soil type which can also impact soil moisture availability.  Are you assuming the soil types are the same under the three vegetation communities and has no effect? 

The text needs extensive review and refinement.  The constant use of prepositions at the beginning of sentences made for a very difficult read and the feeling that many of the sentences were incomplete.  Aside from that, here are a few additional corrections that need to be made:

  • need to be consistent in the use of dataset vs data set and multiyear vs multi-year
  • the section beginning on line 199 should begin with a lead in sentence or two rather than simply starting the section with a bullet
  • line 205 replace ) around 66 with ]
  • correct the spacing in lines 209, 227, 228, and 256
  • Figure 2b: is a better resolution (less blurry) map possible?  Maybe adjust the legend to only include the classes that are present so that colors do not have to be reused. For example, it is currently difficult to tell the difference between water and wetland.
  • Figures 4 and 5 should be swapped to maintain correct chronology (currently Fig 5 is mentioned before Fig 4)
  • Table 2 correct Mean in the 1992 to 2000 column
  • Something seems odd about the way the Figure 6 caption is written in the placement of the (a) and (b).  Please clarify
  • Figure 15 is missing from the manuscript
  • Figure A1 in the appendix is too small to read

Reviewer 3 Report

The main objective of the paper is to monitor the amount of water left in the soil during the harvest named “residual soil moisture” in the presented paper. The work is focused on the autumn soil moisture response to the long-term variability of rainfall derived from CHIRPS product. The variability of soil moisture over the UBNB in Ethiopia was analyzed using a long‐term data set provided by the ESA, CCI product from 1992 to 2017. Also, the evapotranspiration product derived from FLDAS Noah land surface model has been used to understand the effect of ET on autumn soil moisture variability. In general, the paper is well written except some typos “detailed below”. Otherwise, as stated by the authors in the introduction, many efforts have been recently devoted (and undergoing) to investigate the impact of soil moisture on the long term variability of rainfall. However, the contribution of this research is not clear. Furthermore, there is a lack of interpretations of the results. A discussion of a lot of results is needed. For example, how do you explain the negative correlation between ASM and summer rainfall? Also, provide some explanation about why the correlation between ASM and rainfall for land cover changes over seasons. In addition, how do you explain that the spring rainfall impacts the variability of Autumn soil moisture more than Autumn rainfall (Table3)? I don’t know if there are the same irrigated sites over the study area or not? If yes, showing the correlation between rainfall and the ASM variability over rainfed and irrigated fields separately could explain the effect of irrigation on the SM variability. Because Irrigation affects directly soil moisture over agricultural areas.

Moreover, other corrections, most of which related to the language, are following suggested in order to improve the quality of the manuscript.

Line 18: Change “from 1992-2017” to “from 1992 to 2017”

Line 119: Use either “rain-fed” or “rainfed” all over the manuscript, not both

Line 164: add in figure 2 or figure1 the locations of the gauge stations used for the validation of CHIRPS rainfall.

Line 182: Delete one “s” in Methodss

Line 196: Is it a linear average?

Line 209: Change capital letter “Where” to minuscule “where” (all over the manuscript after the equations, e.g: Lines: 215, 234, 238 …)

Equation 2: Define “s” variable

Lines 276-277: Why using just two transects? It’s better to see the effect of topography all over the study area. The two transect doesn’t represent the whole area.

Line 282: Use either “Multiyear” or “Multi-year” all over the manuscript, not both.

Lines 282-283: Figure 4 should appear before figure 5 in the manuscript. Add a phrase presenting figure 4 between figure 3 and Figure 5.

Line 317: Expose 3 in (m3/m3)

Equation 4: “xi” not defined.

Table 2: For the first decade, how do you explain the high variability of ASM over shrubland than forest and agricultural areas?

Figure 6: Change figure caption by:” (a) The monotonic trends (m3/m3 year-1 for ASM) and (b) pixel having significant (p < 0.05) increase or decrease trend in soil moisture over the Upper Blue Nile Basin (UBNB) in Ethiopia for the period of 1992-2017. The positive and negative values indicate the wetting and drying trends of soil moisture in the UBNB, respectively”. In order to be coherent with the other figures.

Line 350: Correct 1992-2017, “2” is missing.

Line 390: Change to “correlation coefficient between autumn soil moisture and rainfall”. Because in this section you indicate the relation between just Autumn SM and Rainfall.

Lines 402-403: Evapotranspiration may explain the negative correlation between ASM and summer rainfall, but looking to the Figure (1), I’m not sure that in this case, the surface runoff is one of the causes to the negative correlation. because surface runoff flux occurs significantly over areas with a high slope. If it is, high elevation pixel (East and center) will show a negative correlation, by contrast, the pixel with low elevation (West) will show a positive correlation. Therefore, could you provide more explanations about the negative correlation between ASM and summer rainfall? Can you add a slope map to see the impact of slope through runoff flux of ASM variability?

Figure 11: It is clear that the lowest correlation between annual rainfall and autumn soil moisture occurred over the peak areas. But in figure 11a, a poor correlation also occurs over low peaks.  For example, in 9.38 degree, that present a low correlation. How this could be explained?

Line 438: Correct “cver” by “cover”.

Line 449: Delete “l” in dynamicsl.

Line 449: Change varaibility by variability.

Table 4:  The correlation between the ASM and rainfall anomalies using FLDAS soil moisture at different depths remain the same in autumn (R=0.91) and doesn’t change a lot for other seasons. How could you explain that? Did the sensing depth doesn’t affect the correlation?

Line 462: “weighted average…” can you explain more how the soil moistures at different depths were calculated? Is it a linear average or not?

Line 478: In figure 12 caption, it is not clear  “Figure (7.11e–h) same as Figures 7.11a–d”  where are these figures (7.11e-h) do you mean ”Figures (e-h) same as Figures (a-d)”.

Figure 12: How does evapotranspiration is more correlated to 0-10 cm Noah soil moisture than ESA CCI SM which represent soil moisture at a thinner layer (0-5 cm)? is the sensing depth does not affect the results?

Line 515: figure 15 is missing.

The clarity of the methodology and results section can be significantly improved by inserting a glossary of symbols and abbreviations used in the manuscript. 

Based on the above considerations, I think the manuscript should be significantly improved and the aim of the work should be better justified in order to warrant his publication.

Round 2

Reviewer 1 Report

The authors has made some modifications to the manuscript, and it seems to have been improved from the previous version, such as the quality of the figures has been improved. Nonetheless, the author did not make modifications for some key issues. The study area is so small that the results obtained may be unstable. In other words, the regionality of the study will limits readers' interest in the research.

Author Response

Thank you

Reviewer 3 Report

The revision addressed the questions and suggested editorial corrections satisfactorily

Author Response

The revision addressed the questions and suggested editorial corrections satisfactorily

This manuscript is a resubmission of an earlier submission. The following is a list of the peer review reports and author responses from that submission.

Round 1

Reviewer 1 Report

The surface soil water content has a low time correlation and therefore the information content of a correlation with seasonal rainfall has a low interest.

The coefficient of variation of soil water content is to be studied in more deatil, non as a whole.

Also, the paper is not well written.